# Efficient Synthesis of Imine-Carboxylic Acid Functionalized Compounds: Single Crystal, Hirshfeld Surface and Quantum Chemical Exploration

**DOI:** 10.3390/molecules28072967

**Published:** 2023-03-27

**Authors:** Muhammad Nawaz Tahir, Akbar Ali, Muhammad Khalid, Muhammad Ashfaq, Mubashir Naveed, Shahzad Murtaza, Iqra Shafiq, Muhammad Adnan Asghar, Raha Orfali, Shagufta Perveen

**Affiliations:** 1Department of Physics, University of Sargodha, Sargodha 40100, Pakistan; 2Department of Chemistry, Government College University Faisalabad, Faisalabad 38000, Pakistan; 3Institute of Chemistry, Khwaja Fareed University of Engineering & Information Technology, Rahim Yar Khan 64200, Pakistan; 4Centre for Theoretical and Computational Research, Khwaja Fareed University of Engineering & Information Technology, Rahim Yar Khan 64200, Pakistan; 5Department of Chemistry, Division of Science and Technology, University of Education, Lahore 54770, Pakistan; 6Department of Pharmacognosy, College of Pharmacy, King Saud University, P.O. Box 2457, Riyadh 11451, Saudi Arabia; 7Department of Chemistry, School of Computer, Mathematical and Natural Sciences, Morgan State University, Baltimore, MD 21251, USA

**Keywords:** single-crystal XRD, condensation reaction, Hirshfeld surface analysis, density functional theory, NBO analysis

## Abstract

Two aminobenzoic acid based crystalline imines (**HMBA** and **DHBA**) were synthesized through a condensation reaction of 4-aminobenzoic acid and substituted benzaldehydes. Single-crystal X-ray diffraction was employed for the determination of structures of prepared Schiff bases. The stability of super molecular structures of both molecules was achieved by intramolecular H-bonding accompanied by strong, as well as comparatively weak, intermolecular attractive forces. The comparative analysis of the non-covalent forces in **HMBA** and **DHBA** was performed by Hirshfeld surface analysis and an interaction energy study between the molecular pairs. Along with the synthesis, quantum chemical calculations were also accomplished at M06/6-311G (d, p) functional of density functional theory (DFT). The frontier molecular orbitals (FMOs), molecular electrostatic potential (MEP), natural bond orbitals (NBOs), global reactivity parameters (GRPs) and natural population (NPA) analyses were also carried out. The findings of FMOs found that *E*_gap_ for **HMBA** was examined to be smaller (3.477 eV) than that of **DHBA** (3.7933 eV), which indicated a greater charge transference rate in **HMBA**. Further, the NBO analysis showed the efficient intramolecular charge transfer (ICT), as studied by Hirshfeld surface analysis.

## 1. Introduction

In the modern world, scientists are trying to explore every possible aspect of newly synthesized chemical building blocks in order to find their maximum utility in the betterment of humanity. In this scenario, crystalline organic compounds are also considered to be important chemical building blocks as their synthesis and computational examination are substantially recognized in modern times. Imines (also called Schiff bases) are a significant class of chromophores produced by the condensation reaction of primary amines and carbonyl chromophores (ketones or aldehydes), causing azomethine (–C=N–) formation [1]. Imines are vital class of organic compounds with numerous applications in the field of coordination chemistry as the azomethine (–C=N–) group is considered as the prized ligand for the production of numerous metal complexes of valuable interest [2,3]. Besides coordination chemistry, Schiff bases are strong candidates in analytical chemistry because of their significant applications in this field [4]. Moreover, the exploration of Imines in the field of medicinal chemistry has also been well recognized because of their potential as anticancer [5], anti-tubercular [6], analgesic and anti-inflammatory [7], antimicrobial [8,9], antioxidant [10,11], anthelmintic [12] and anticonvulsant medicines [13,14,15,16,17], etc. Imines derived from primary amines are basic in nature, while in acidic solution they readily convert in to iminium ions [18,19]. Iminium catalysis for the asymmetric organic transformation is another area where they are well recognized [20]. Iminium ion/imines/Schiff bases are the key intermediates in multi-component reactions (Ugi four-component procedure) as well as having remarkable roles in fabrication of alkaloids and various nitrogen-containing heterocyclic compounds [21,22,23]. One of the very important features of this functionality is its existence as a zwitterion or inner salt that contributes significantly towards the modification of physical properties, such as solubility enhancement and boasting permeability of the compounds; these are extremely appreciated features in the field of designing drugs [24]. On the other hand, carboxylic acids are a significant class of organic systems with multiple applications in the fields of agriculture, medicine, food, pharmaceuticals and other industries [25]. These organic acids and their derivatives are used in the production of biopolymers, polymers, coatings, pharmaceutical drugs, adhesives, solvents, antimicrobials, food additives, flavorings, etc. [26]. This is also a crucial class of organic chromophores with a number of advantages in the coordination chemistry arena, as the carboxylate functional moiety has been well analyzed as a valuable ligand for the production of numerous metal complexes of valuable interest [26,27,28,29]. The overlapping of discrete research grounds for the exploration, reasoning and investigation of the major electronic characteristics of the yielded organic systems is a recently acknowledged research area. In this context, density functional theory computations are greatly recognized as a significant method for the assessment of basic electronic characteristics of the recently manufactured chromophores, and, likewise, non-covalent interactions and nonlinear optical properties [30,31,32]. Functionalized azomethine have also been investigated by the DFT analysis for the detailed investigation of their key electronic properties [33].

Duel functionalized compounds such as 4-amino benzoic acid are excellent candidates for modification to valuable chemical architectures that, in turn, could be utilized for developing stabilized silver nanoparticles with potential analytical probes for the selective colorimetric sensing of Ni^2+^ ions and antimicrobial performance [34]. As our group is continuously working on the synthesis and DFT-based exploration of organic compounds belonging to different classes, here, we are showcasing our research relating to the synthesis, single-crystal investigation and DFT study of the carboxylic acid functionalized imine compounds: **DHBA** and **HMBA**. Furthermore, the free carboxylic acid functionality will further be applied for the production of valuable coordination compounds [35].

## 2. Result and Discussion

### 2.1. Description of SC-XRD Analysis of HMBA and DHBA

The crystal structure and the mode by which the molecules are packed in **HMBA** and **DHBA** are elaborated in a detailed way. The details related to the SC-XRD experiment are depicted in Appendix A (Appendix A) specify the bond lengths as well as the bond angles of **HMBA** and **DHBA** compared with the results of DFT.

The crystalline structure of **HMBA** has a 2-{(E)-[(4-carboxyphenyl)iminio]methyl}-6-methoxyphenolate (C1-C15/N1/O1-O4) and a methanol molecule (Figure 1a), whereas **DHBA** contains a (E)-4-((2-hydroxy-3,5-diiodobenzylidene)amino)benzoic acid (C1/C2A-C7A/C8-C14/N1/O1-O3/I1/I2) and a dimethyl sulfoxide (DMSO) molecule (Figure 1b). 2-{(E)-[(4-carboxyphenyl)iminio]methyl}-6-methoxyphenolate adopts the keto tautomeric form, whereas (E)-4-((2-hydroxy-3,5-diiodobenzylidene)amino)benzoic acid adopts the enol tautomeric form. In **HMBA,** the ring A (C1-C7/N1/O1/O2) and group B (C8-C15/O3/O4) are roughly planar with a root mean square (r.m.s) deviation of 0.0800 and 0.0115 Å, with A/B dihedral angle of 8.7 (2)°. In **DHBA**, the phenyl ring (C2A-C7A) is refined as disordered over two positions by employing the equal anisotropic parameters of all the atoms involved in disorder, and the refined occupancy ratio is 0.538 (13):0.462 (13). In **DHBA**, the similar ring A (C1/C2A-C7A/N1/O1/O2) and group C (C8-C14/O3/I1/I2) are roughly planar with r.m.s deviations of 0.1116 and 0.0446 Å. The carboxylate group oxygen atom (O2) of group C showed maximum deviation from the plane with a deviation of 0.2322 (9) Å. The intramolecular N-H⋯O and O-H⋯N bonding are the primary characteristic of the molecular configuration of **HMBA** and **DHBA,** respectively. By the intramolecular H-bonding in **HMBA** and **DHBA**, the S (6) H-bonded loop is formed. There is some sort of similarity in the mode by which compounds are packed in both compounds. In both compounds, the main molecule is associated with the solvent molecule through O-H⋯O bonding. In **HMBA**, the 2-{(E)-[(4-carboxyphenyl) iminio]methyl}-6-methoxyphenolate molecules are directly interconnected via C-H⋯O bonding; whereas, in **DHBA**, the (E)-4-((2-hydroxy-3,5-diiodobenzylidene)amino)benzoic acid molecules are not interlinked directly with each other via any sort of H-bonding, but these are linked through the DMSO molecule. In **HMBA**, the molecules are connected in the form trimer through O-H⋯O and C-H⋯O bonding to complete the $R_{3}^{3}\left(21\right)$ loop [36]. The trimers are further interlinked with each other through C-H⋯O bonding. The C (11) chains of molecules are formed via O-H⋯O and C-H⋯O bonding that run along the *b*-axis (Figure 2, Table 1). Similarly, in **DHBA**, the chains of molecules are formed via intermolecular O-H⋯O and C-H⋯O bonding (Appendix A and Table 1). The weak π⋯π stacking interactions among aromatic rings cause further strengthening of the crystalline packing of **HMBA** via inter-centroid separation of 4.600 (5) to 5.198 (5) Å; whereas, the C-I⋯π interaction with I⋯π distance of 3.783 Å (Figure 3) and π⋯π stacking interaction via inter-centroid separation of 4.328 (6) to 5.849 (6) Å are responsible for the further strengthening of the crystal packing of **DHBA**. Moreover, a Cambridge structure database search was performed in order to find the closely related compounds. A general search of (E)-(4-(benzylideneamino) phenyl) methanol provides a lot of hints. Out of these hints, some very closely related compounds are found with the reference codes BZANPC10 [37], BZANPC11 [38], DIWMAY [39], FAXXIN [40], NEZXAT [41], PUSMUN [42]. A comparative study on geometrical parameters between DFT and XRD was also accomplished, and harmony between the values was examined (Appendix A and Appendix A) which indicates the suitable selection of DFT functional for the current study.

### 2.2. Hirshfeld Surface Analysis

In the field of supramolecular chemistry, the need for the detailed survey of the non-covalent interactions is increasing day by day as it forms the basis of crystal packing. Herein, we are focused on exploring these interactions in the studied compounds via Hirshfeld surface analysis through Crystal Explorer v. 21.5 [43]. A Hirshfeld surface (HS), which is constructed on the basis on normalized distances ($d_{\mathrm{norm}}$), enables us to explore the hydrogen bonding interactions [31,44,45,46,47]. Red regions on the surface stand for the smaller interatomic contact, whereas blue parts stand for larger interatomic contacts. The region of white color of HS stands for the contacts in which the distance among the associated atoms is equal to the sum of the van der Waal radii of atoms. The red spots on HS for **HMBA** (Figure 3a) and **DHBA** (Figure 3b) indicate atoms that are involved in small interatomic contacts or in H-bonding interactions. Shape index is the property that is employed for plotting HS to explore π⋯π interactions. The regions of red and blue having the shape of a triangle around the aromatic rings on HS (shape index) are the indicators of the π⋯π stacking interactions in **HMBA** (Figure 3c) and **DHBA** (Figure 3d).

The vital or precious interatomic contacts present in the crystal packing of single-crystals can be effectively explored by employing 2D finger print plots [48,49]. The plots for whole interactions are represented by Appendix A for **HMBA** and Appendix A for **DHBA**. The largest spikes on these plots stand for O⋯H contact. A significant difference is found in the interatomic contacts of **HMBA** and **DHBA**. For **HMBA**, the contact of utmost important is H⋯H (42.2%, Appendix A), whereas for **DHBA**, the contact of utmost importance is O⋯H (24.3%, Appendix A). The C⋯H and O⋯H contacts have substantial contributions in the crystal packing of both compounds which is the evidence of the C-H⋯O bonding in them. The other interatomic contacts for **HMBA** and **DHBA** are shown by Appendix A and Appendix A, respectively. In order to make the study of short contacts more interesting, we computed the enrichment ratio for finding the tendency of the pair of the chemical species to form crystal packing associations for both compounds. The enrichment ratio is calculated by following the procedure created by the Christian Jelsch et al. [50]. The results of the enrichment ratio are exhibited in Appendix A for **HMBA** and Appendix A for **DHBA**. For **HMBA**, the contacts that are more favorable than other contacts are C⋯C, O⋯H/H⋯O with enrichment ratios of 2.52 and 1.43, respectively. For **DHBA**, the contacts that are more favorable than other contacts are I⋯I, S⋯H/H⋯S, C⋯C with enrichment ratios of 2.27, 1.92 and 1.48, respectively. C⋯C contacts are favorable contacts in both compounds as both compounds contain aromatic rings.

In the exploration of the single crystals, the scientist tries to determine the strength of crystal packing. We also do the same by exploring the void analysis [51,52] of **HMBA** (Figure 4a) and **DHBA** (Figure 4b). All the atoms are supposed to be spherically symmetrical and the electron density of all the atoms is added up in order to find the voids. Overall, 148.02 Å^3^ and 216.36 Å^3^ are the calculated voids in **HMBA** and **DHBA**, respectively. The void in **HMBA** is 9.98% and in **DHBA** it is 11.4% which shows the absence of any sort of cavity in the crystalline packing of both compounds.

In order to further investigate the interactions in **HMBA** and **DHBA**, interaction energy between the main molecule and the solvent molecule is calculated on crystal explorer using the HF/3-21G electron density model. The intermolecular interaction energy is the sum of four kinds of energies named as electrostatic, dispersion, polarization and repulsion. The electrostatic energy can be attracted or repulsive. The strength of the interaction can be shown by the width of the cylinder connected with the center of the molecules. A larger width of the cylinder indicates a stronger interaction. In the case of **HMBA**, the roles of the electrostatic and dispersion energies in defining the total energy are almost the same. The contribution of Coulomb electrostatic energy is a little bit larger (−8.2 kJ/mole, Figure 5a) as compared to dispersion energy (7.3 kJ/mole, Figure 5b). In the case of **DHBA**, the dispersion energy is far higher than the Coulomb electrostatic energy. The contribution of dispersion energy is −24.2 kJ/mole (Figure 5e), whereas the contribution of coulomb electrostatic energy is just −2.9 kJ/mole (Figure 5d). The total interaction energy is attractive for both compounds, but this energy is larger in **DHBA** as compared to **HMBA**.

### 2.3. Natural Population and Molecular Electrostatic Potential Analyses

A brief investigation on the atomic charges derived from natural bond orbitals (NBOs) of compounds **HMBA** and **DHBA** was presented in Appendix A. This approach provides a good input for obtaining information about the polarizability of molecules and electronic properties of the chemical systems [53]. The phenomenon associated with the charge transformation originating in reactions, the electrostatic potential on exteriors of crystal and the electronegativity equalization are frequently evaluated using this analysis [54]. The charges observed on atoms are considered as a substantial factor for bonding capacity as well as molecular conformation. The unequal electron density redistribution data, indicated by NBO charges, of studied compounds reveals the presence of highly electronegative elements such as nitrogen and oxygen [55]. Moreover, the MEP plot could be utilized to predict the chemical nature and reaction sites in the studied compounds [56,57]. Accompanying NPA, the MEP descriptor could be regarded as significant for understanding the EP surface activity and non-covalent interactions (NCIs) in the aforementioned compounds. The MEP surface distinguishes the reaction sites of any compound via various standard colors, such as orange, red, yellow, blue and green, to comprehensively explain the regions of electrophilic as well as nucleophilic attack. The charge contribution systematically declines in the order of red > orange > yellow > green > blue, where deep red specifies negative potential and deep blue represents positive potential [58]. In **HMBA** and **DHBA**, the electronic charges based on NPA and MEP heat maps were estimated by adopting M06 as a level of theory. NPA graphs of **HMBA** displayed (Appendix A) that some carbon and most of the hydrogen atoms were positively charged, which was further supported by MEP maps (Appendix A) accomplished at the M06/6-31G(d,p) functional of DT-DFT, represented by a blue color. As the blue colored segment represents the nucleophile enticing site, it is assured to communicate with the most positive potential. Whereas, most of the C, N and O atoms were negatively charged, which is favorably prone to electrophilic attack as indicated by a red color (Appendix A). Similarly, for **DHBA**, iodine, sulfur, some of the carbons and most of the hydrogens were seen as positively charged. All the atoms were found positively charged, while oxygen, nitrogen and most of the carbons were negatively charged (See Appendix A); this was further supported by MEP maps with blue and red colors, respectively, as shown in Appendix A. Additionally, the plotted property dnorm of HS analysis also supported the NPA charges as the more electronegative regions (N, O and I) in HS are a red color, while electrophilic sites (C and H) are highlighted with blue and yellow colors (Figure 3). The development of NCI (S⋯H/H⋯S and O⋯H/H⋯O) in molecules supported the NPA charges.

### 2.4. NBO Analysis

The NBO analysis provides a clear understanding regarding the molecular interactions, hyperconjugation and the charge transfer phenomena in the acceptor (Lewis acid) and donor (Lewis base) moieties [59,60]. Strong interactions between the donor and acceptor moieties harvest a large value for stabilization energy. The energy of stabilization *E*^(2)^ paired with the delocalization for every donor (*i*) and acceptor (*j*) moiety was estimated using Equation (1).(1)$$E^{\left(2\right)}=q_{i}\frac{{\left(F_{i,j}\right)}^{2}}{\varepsilon_{j}-\varepsilon_{i}}$$
where *E*^(2)^ represents stabilization energy, the diagonal = *F*(*i*,*j*), donor orbital occupancy = *q_i_*, and the off-diagonal elements = *ε_i_*, *ε_j_* are NBO Fock matrix elements. Some important outcomes of NBOs for compounds **HMBA** and **DHBA** are listed in Table 2; meanwhile, Appendix A demonstrated the rest of the transitions, representing established conjugation.

The σ→σ*, π→π*, LP→σ* and LP→π* overlaps resulted in the intramolecular hyper conjugative interactions, which facilitated ICT, consequently resulting in the stability of the system. NBO analysis was performed for **HMBA** and **DHBA** at M06/6-311 G(d,p) functional. The comprehensive detail for NBO analysis of **HMBA** and **DHBA** was provided in Appendix A and some typical values for **HMBA** and **DHBA** were presented in Table 2. In the studied compounds, the following transitions: σ→σ*, π→π*, LP→σ* and LP→π* are recognized. Among σ→σ*, important electronic transitions, σ (C8-H20)→σ* (C9-C10) and σ (C9-C10)→σ* (C10-C11) were spotted with the maximum stabilization energies, i.e., 5.15 and 6.42 kcal/mol for **HMBA** and **DHBA** compounds, respectively. Meanwhile, the transitions such as σ (C15-H31)→σ* (O4-C11) and σ (O3-H7)→σ* (N1—C5) were seen with the lowermost stabilization energies, i.e., 0.55 and 0.52 kcal/mol of **HMBA** and **DHBA** compounds, respectively. In case of compound **HMBA**, π→π* interactions such as π (C4-C5)→π* (C2-C3) having maximum stabilization energy values of 26.32 kcal/mol and π(C2-C3)→π* (C2-C3) having minimum stabilization energy value of 0.61 kcal/mol were seen. In compound **DHBA**, among π→π* transitions, π (C10-C11)→π* (C12-C13) and π (C9-C14)→π* (C9-C14) were significant transitions with maximum and minimum stabilization energies, i.e., 30.68 and 0.67 kcal/mol, respectively. Among LP→σ* transitions, LP2(O1)→σ* (O2−C1) and LP2(O1)→σ* (O2−C1) were found as the prominent transitions with the stabilization energies of 34.41 and 34.35 kcal/mol for **HMBA** and **DHBA** compounds, respectively. Meanwhile, LP→π* prominent transitions such as LP(2) (O2)→π* (O2-C1) and LP(2) (O2)→π* (O1−C1) were obtained with the stabilization energies of 47.09 and 47.53 kcal/mol for **HMBA** and **DHBA** compounds, respectively. The stability of studied compounds was credited by good stabilization energies due to the phenomenon of resonance. The existence of hyperconjugation, extended conjugation and charge transfer were conclusively and precisely confirmed by the NBO data. However, the synergistic effect of charge transfer caused by extended conjugation leads to a significant red shifted behavior among the investigated compounds.

### 2.5. FMO Analysis

Through frontier molecular orbitals (FMOs) investigation, entitled compounds were not only analyzed for chemical stability but also interpreted for the optical as well as electronic responses [61,62]. HOMO to LUMO energy variation was considered to be responsible for the stability and reactivity of a compound. [63,64]. Furthermore, this energy difference (∆*E* = *E_LUMO_* − *E_HOMO_*) was used for computing the various GRPs of the synthesized molecules (see Table 3). A greater chance of ICT was provided with a smaller energy gap among LUMO and HOMO orbitals and based on how effectively polarizable the molecule was and vice-versa [65,66]. The electronic properties of **HMBA** and **DHBA** compounds were determined by FMOs at the M06 level of TD-DFT. The energies of the first, second and third excited states of FMO orbitals were presented in Table 3.

The band gaps of HOMO/LUMO in **HMBA** and **DHBA** compounds were found as 3.477 and 3.793 eV, respectively, with −5.894 and −6.533 eV. The energy gap noted in **DHBA** is higher as compared to **HMBA** which indicated it as more stable, and hard with little reactivity. The lower band gap in **HMBA** elucidated it as more reactive, less stable and softer than the former compound. This difference might originate from the presence of the electronegative iodine atom. In order to check the coherence in the energies, we calculated the energy level up to HOMO-2/LUMO+2. The *E*_gap_ of HOMO-1/LUMO+1 in **HMBA** and **DHBA** was observed as 5.891 and 5.181 eV, correspondingly. Likewise, the HOMO-2/LUMO+2 energy gap of **HMBA** and **DHBA** was noticed as 6.489 and 6.001 eV, respectively. Accompanying the energies of orbitals, the charge transference between the orbitals can also be estimated through FMOs study (see Figure 6). In the case of compound **HMBA**, in HOMO, the charge density was present on the part (Z)-2-methoxy-6-((methyl- λ^2^-azaneyl)methylene)cyclohexa-2,4-diene-1-one; whereas, in LUMO, the electronic cloud was dispersed over the entire compound. In addition, in **DHBA**, for HOMO, the charge density is primarily located on the dimethyl sulfoxide group of the crystal; whereas, in context of LUMO, it is condensed on the complete molecule. For HOMO-1/LUMO+1 and HOMO-2/LUMO+2, the electron density is located over the molecule in **HMBA**. Similarly, a trend is seen for **DHBA** in the case of HOMO-2/LUMO+2, while for HOMO-1/LUMO+1, the electronic cloud is located at (Z)-2-methoxy-6-((methyl-λ^2^-azaneyl)methylene)cyclohexa-2,4-diene-1-one.

### 2.6. Global Reactivity Parameters

The electronegativity (*X*) [67], electron affinity (*A*), ionization potential (*I*) [68], global electrophilicity index (*ω*) [69], global softness (*σ*) [70], chemical potential (*μ*) [71] and global hardness (*η*) [72] were calculated by utilizing the energies of FMOs. The GRPs of **HMBA** and **DHBA** compounds are estimated by utilizing Equations (2)–(7) and outcomes are illustrated in Table 4.

The major findings obtained via Equations (6)−(11) were shown in Table 4. Thus, the electron withdrawing and donating capabilities of the aforementioned compounds were depicted by *I* and *A*. In the aforementioned compounds, the values of *I* and *A* were found higher in **DHBA** as compared to **HMBA**, which showed that **DHBA** had a greater ability to accept the electron than **HMBA**. Moreover, the electronegativity values were found to be [**DHBA** (*X* = 4.636 eV)] > [**HMBA** (*X* = 4.155 eV)]. The global electrophilicity (*ω*) trend for **HMBA** and **DHBA** was 4.966 and 5.667 eV, respectively. The reactivity as well stability of the studied chromophores are indicated by the parameters such as global softness (*σ*) and global hardness (*η*) of the studied compounds. The molecules possessing a smaller band gap could be considered as soft, chemically reactive and unstable. Contrarily, compounds with a larger band gap can be categorized as hard, less reactive and kinetically stable candidates. Table 4 showed that the compound HMBA had a higher global softness value (*σ* = 3.484 eV^−1^) than **DHBA** (*σ* = 3.802 eV^−1^). Similarly, the value of global hardness for **DHBA** was higher than **HMBA**, i.e., 5.667 and 4.966, respectively. All the global reactivity parameters collectively disclosed that **HMBA** was a softer and more reactive compound than **DHBA** (Table 4).

### 2.7. Dipole Moment and Liner Polarizability

Inorganic and organic materials are regarded as effective optical materials because of their significant contributions in optical logic for advanced technologies such as signal modulation and telecommunications [73,74]. The intra-molecular charge transfer and larger conjugation of electrons in organic chromophores have expanded their roles as opto-electronic materials as compared to inorganic compounds [68,69,70,71,72,73,74,75]. The values of dipole moment (*µ*) and linear polarizability <*α*> for studied chromophores were computed by applying M06 level of theory; the results were tabulated in Appendix A. Urea is the reference compound most commonly exploited in the examination of the opto-electronic response of various molecules, and is considered as a threshold standard for comparing the <*α*> response of studied molecules [76]. **HMBA** and **DHBA** are regarded as polar molecules as they comprise non-zero dipole moments 6.34 and 5.76 D, respectively. The higher contribution among all the axes is seen along the y axis: µ_y_ = −5.55 and 5.68 D in **HMBA** and **DHBA**, respectively. The values of dipole moment calculated for the studied molecules are found to be greater as compared to urea (1.37 D) [77]. The computed average polarizabilities <*α*> of **HMBA** and **DHBA** were found out to be 313.48 and 288.79 a.u, respectively. Among explored molecules, the average polarizability value of **HMBA** is greater compared to **DHBA**. These values of average polarizability <*α*> of **HMBA** and **DHBA** are found to be 11.34 and 10.45 times larger than standard molecule urea (<*α*> = 27.63) [25]. The major contribution tensor is examined along the x-axis ***α_xx_*** = 528.79 and 436.95 a.u, for **HMBA** and **DHBA**, respectively, among all the tensors.

## 3. Methodology

### 3.1. Experimental Details

The carboxylic acid functionalized imines compounds were synthesized utilizing 4-aminobenzoic acid and substituted benzaldehydes of maximum purity, as they were purchased and used without further purification. The standard chemicals were purchased from the reputed chemical suppliers such as Sigma-Aldrich, Macklin, Acros Chemicals, BDH and TCI. Solvent purification was accomplished by a simple distillation procedure. Thin-layer chromatography (TLC) employing the aluminum sheet priorly coated with silica gel was utilized for the reaction progress. For the SC-XRD study, a Bruker Kappa Apex-II diffractometer was utilized with an X-ray source producing Mo-Kα radiations. Software Apex-II [78], SAINT [79], SHEXS-97 [80] and SHEXL 2018/3 [81] were used for the sake of collecting data, integrating data, making a structure solution from raw data, and for refining data, respectively. All the H-atoms of **HMBA** and **DHBA** were assigned relative isotropic displacement parameters except the H-atom attached to N1 and O2 of **HMBA** which were refined independently by keen investigation of the remaining electron density peaks. The SC-XRD findings were graphically represented by PLATON [82], Mercury 4.0 [83] and ORTEP-3 [84] software.

### 3.2. Synthesis of the Carboxylic Acid Functionalized Imines

The carboxylic acid functionalized imines compounds **HMBA** and **DHBA** were produced by the condensation reaction strategy using 4-aminobenzoic acid and substituted benzaldehyde.

#### 3.2.1. Synthesis of (E)-4-((3,5-Diiodo-2-Hydroxybenzylidene)Amino)Benzoic Acid (DHBA)

In a round bottom flask (50 mL), we had 3,5-diiodo-2-hydroxybenzaldehyde (1 mmol) in 30 mL methanol to which we added 4-aminobenzoic acid (1.2 mmol) and the reaction mixture was subjected to reflux conditions for 3 h. After completing the reaction (guessed by TLC), methanol was removed under lower pressure and then an aqueous workup of the mixture was performed. The dry organic layer was reduced using a rotary evaporator. The compound was purified using column chromatography and recrystallized for the SC-XRD analysis.

#### 3.2.2. Synthesis of (E)-4-((2-Hydroxy-3-Methoxybenzylidene)Amino)Benzoic Acid (HMBA)

The synthesis of chromophore **HMBA** was also carried out by utilizing the same procedure as above, except using 2-hydroxy-3-methoxybenzaldehyde as the aldehyde component and methanol as the solvent, see Figure 1.

### 3.3. Computational Procedure

In the current study, all the DFT calculations for aminobenzoic acid-based compounds (**HMBA** and **DHBA**) were accomplished through the Gaussian 09 program [85] at M06/6-311G(d,p) functional [86]. From the literature, we found that M06/6-31G (d,p) is highly parameterized to approximate exchange-correlation energy functionals in DFT, utilized significantly for calculations of excited states, non-covalent interactions and bonding phenomena; therefore, we utilized M06 functional for this investigation [86,87]. At first, optimizations of geometries were performed, and confirmation of the true minima was determined by the lack of negative frequencies. Various kinds of analyses, NBO, NPA, dipole moment and linear polarizability analyses were performed by utilizing the optimized geometries of **HMBA** and **DHBA** in the gaseous phase. For the NBO investigation, NBO software package 3.1 [88] was utilized. Time-dependent density functional theory (TDDFT) was used to calculate the FMOs, and MEP-based findings. To understand the reactivity of the aforementioned compounds, GRPs were determined by utilizing energies of HOMOs/LUMOs with the aid of Equations (2)–(7).
*I* = −*E*_HOMO_(2)
*A*= −*E*_LUMO_(3)
(4)$$\eta =\frac{I-A}{2}$$
(5)$$X=\frac{I+A}{2}$$
(6)$$\omega =\frac{\mu 2}{\eta }$$
(7)$$\sigma =\frac{1}{2\eta \;}$$

Different types of software such as Gauss View 5.0 [89], Gauss Sum [90], Chemcraft [91] and Avogadro [92] had been utilized for the interpretation of data from outputs.

## 4. Conclusions

The synthesis of two carboxylic acid functionalized imines compounds **DHBA** and **HMBA** has been accomplished in reasonable yields and characterized by SC-XRD analysis. This study specifies the effect of intramolecular interaction in the stabilization of the molecular configuration of both compounds and the roles of strong and weak NCIs in the packing of the crystals. The intermolecular interactions of strong and weak natures are explored and compared by Hirshfeld surface analysis. The synthesized crystals are computed for their electronic, structural and comparative analyses. The NBO study depicted that the intramolecular H-bonding and some prominent transitions LP2(O1)→σ*(O2−C1) and LP2(O1)→σ*(O2−C1) with the highest energy of stabilization of 34.41 and 34.35 kcal/mol for **HMBA** and **DHBA**, respectively, exist in both synthesized compounds, and this played an important role in stabilizing the compounds. The FMOs analysis revealed that the band gap (3.477 eV) of **HMBA** is comparatively smaller than that of **DHBA** (3.793 eV), which shows higher polarizability in **HMBA**. The comparative study with urea indicated that both molecules exhibited significant behaviors of linear responses and can be used as efficient opto-electronic materials.

**X-ray Crystallography**: Deposition Numbers 2117442 for **HMBA** and 2117443 for **DHBA** contain the supplementary crystallographic data for this paper. These data are provided free of charge by the joint Cambridge Crystallographic Data Centre and Fachinformationszentrum Karlsruhe Access Structures service.

## Data Availability

The experimental details including computational details are given in the supporting information.

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
