# Peer review of "Efficient Synthesis of Imine-Carboxylic Acid Functionalized Compounds: Single Crystal, Hirshfeld Surface and Quantum Chemical Exploration"

_molecules, 2023, doi:10.3390/molecules28072967_

Round 1
Reviewer 1 Report
The comments have been included in a pdf file

Author Response
Answer to the reviewer's comments to the author:
Reviewer #1:
Comment to Author:
Query 1. Source and purity of all chemicals used should be specified in the experimental section.
Response: As suggested, source and purity of all chemicals used have been specified in the experimental section of the revised manuscript and highlighted in yellow.
Query 2. Please provide the H and 13C spectra and MS data.
Response: We are thankful to the respected reviewer for comments. Regarding the spectroscopic characterization, with due respect, at the time of experimental work, we analyzed and explored the crystal of the product by SC-XRD technique that fully confirmed the expected structure. Currently, it is very difficult to further characterize the compound by NMR technique as we do not have NMR facilities at our university.. We do think that we have analyzed the structure by single crystal analysis and discussed in very much detail for the better understanding without any ambiguity and hope will be enough for the readers. We hope that the respected reviewer will allow us to justify the structures by the single crystal analysis as well as DFT based study.
Query 3. The Fig. 2 should be redraw, it is not well. Also the ring and chain have a term in such descriptions, such as R/C etc.
Response: Dear respected Reviewer, we redraw Fig. 2. We carefully analyze the intra and intermolecular interactions in HMBA. The description of the crystal packing is now corrected. Symbols S, R and C are used to represent intramolecular H-bonded loop, intermolecular H-bonded loop and infinite chain, respectively. The main article that shows the representations of H-bonded loops and chains is now cited (Bernstein, Joel, Raymond E. Davis, Liat Shimoni, and Ning‐Leh Chang. "Patterns in hydrogen bonding: functionality and graph set analysis in crystals." Angewandte Chemie International Edition in English 34, no. 15 (1995): 1555-1573). Thanks for you comment.
Query 4. “Moreover, exploration of Imines in the field of medicinal chemistry has also been well recognized because of their potential as anticancer, anti-tubercular, analgesic and anti-inflammatory, antimicrobial, antioxidant, anthelmintic and anticonvulsant medicines etc.” some related documents could be updated, such as Chem. Commun., 2022, 58, 6653–6656; Org. Chem. Front., 2020,7, 3515-3520 and J. Org. Chem. 2019, 84, 14627−14635. “The MEP heat maps of HMBA and DHBA, which represent the 3D spreading of electronic cloud, were carried out using time dependent DFT (TD-DFT) method at M06 level and displayed in Figure S7.” This part should be cited, such as ACS Omega, 2018, 3, 17986−17990, J Comput Chem, 2018, 39, 117–129.
Response: As suggested, all the five article have been cited in the revised manuscript at suggested places as Ref.no 13, 14, 16, 70 and 71
- Wang, Y.-F.; Wang, C.-J.; Feng, Q.-Z.; Zhai, J.-J.; Qi, S.-S.; Zhong, A.-G.; Chu, M.-M.; Xu, D.-Q. Copper-Catalyzed Asymmetric 1, 6-Conjugate Addition of in Situ Generated Para-Quinone Methides with β-Ketoesters. Chem. Commun. 2022, 58, 6653–6656, doi:https://doi.org/10.1039/D2CC00146B.
- Yao, W.; Wang, J.; Zhong, A.; Wang, S.; Shao, Y. Transition-Metal-Free Catalytic Hydroboration Reduction of Amides to Amines. Org. Chem. Front. 2020, 7, 3515–3520, doi:https://doi.org/10.1039/D0QO01092H.
- Yao, W.; He, L.; Han, D.; Zhong, A. Sodium Triethylborohydride-Catalyzed Controlled Reduction of Unactivated Amides to Secondary or Tertiary Amines. J. Org. Chem. 2019, 84, 14627–14635, doi:• DOI: 10.1021/acs.joc.9b02211.
- Zhao, D.; Liu, S.; Rong, C.; Zhong, A.; Liu, S. Toward Understanding the Isomeric Stability of Fullerenes with Density Functional Theory and the Information-Theoretic Approach. ACS Omega 2018, 3, 17986–17990, doi:https://doi.org/10.1021/acsomega.8b02702.
- Cao, X.; Rong, C.; Zhong, A.; Lu, T.; Liu, S. Molecular Acidity: An Accurate Description with Information-Theoretic Approach in Density Functional Reactivity Theory. J. Comput. Chem. 2018, 39, 117–129, doi:https://doi.org/10.1002/jcc.25090.
Query 5. Please check the data in Table 3 and Fig6.
Response: Now in revised manuscript, we critically rechecked the values and made correction. Thanks for the comment.
Query 6. Computational procedure, this part should be illustrated in detail.
Response: Now in revised manuscript, we discussed computational procedure in detail as per the suggestion of reviewer. Thanks for the comments.
Computational procedure
In current study, all the DFT calculation for aminobenzoic acid based compounds (HMBA and DHBA) were accomplished through Gaussian 09 program [43] at M06/6‐311G(d,p) functional [44]. From literature we found that M06 / 6-31G (d, p) is highly parameterized approximate exchange-correlation energy functionals in DFT, utilized significantly for calculations of excited states, non-covalent interactions and bonding phenomena, therefore, we utilized M06 functional for this investigation [44, 45] At first, optimizations of geometries were performed and confirmation of true minima was determined by the lack of negative frequencies. Various kind of analyses: NBO, NPA, dipole moment and linear polarizability analyses were performed by utilizing the optimized geometries of HMBA and DHBA in gaseous phase. For NBO investigation, NBO software package 3.1 [46] was utilized. Time-dependent density functional theory (TDDFT) was used to calculate the FMOs and MEP based findings. To understand the reactivity of afore-said compounds, GRPs were determined by utilizing energies of HOMOs/LUMOs with the aid of Equations 1-6
|
I= -EHOMO |
(1) |
|
A= -ELUMO |
(2) |
|
η = |
(3) |
|
X= |
(4) |
|
ω = |
(5) |
|
σ = |
(6) |
Different softwares such as GaussView 5.0 [47], GaussSum [48], Chemcraft [49]and Avogadro [50] had been utilized for interpretation of data from outputs.
Reviewer 2 Report
1. Source and purity of all chemicals used should be specified in the experimental section.
2. Please provide the H and 13C spectra and MS data.
3. The Fig. 2 should be redraw, it is not well. Also the ring and chain have a term in such descriptions, such as R/C etc.
4“Moreover, exploration of Imines in the field of medicinal chemistry has also been well recognized because of their potential as anticancer, anti-tubercular, analgesic and anti-inflammatory, antimicrobial, antioxidant, anthelmintic and anticonvulsant medicines etc.” some related documents could be updated, such as Chem. Commun., 2022, 58, 6653–6656; Org. Chem. Front., 2020,7, 3515-3520 and J. Org. Chem. 2019, 84, 14627−14635. “The MEP heat maps of HMBA and DHBA, which represent the 3D spreading of electronic cloud, were carried out using time dependent DFT (TD-DFT) method at M06 level and displayed in Figure S7.” This part should be cited, such as ACS Omega, 2018, 3, 17986−17990, J Comput Chem, 2018, 39, 117–129.
5. Please check the data in Table 3 and Fig6.
6. Computational procedure, this part should be illustrated in detail.
Author Response
Comment to Author:
The article is an interesting study, would like to re-review it and probably publishable after considering following points
Response: We are deeply thankful to the respected reviewer for the positive recommendation.
Query 1. The introduction is too general, at least a section dealing with practical applications of compounds of the same or similar type as those synthesised should be included.
Response: Thank for the comments. Actually we have discussed two functionalities i.e imines and carboxylic acid, in the introduction section we have discussed the applications of imines in asymmetric catalysis for organic transformation, medicinal chemistry as well as its applications of imines and carboxylic acid functionalities in coordination chemistry for the synthesis of valuable complexes where the suitable references have been cited.
Query 2. Equations (6) and (7) seem to be changed in relation to the text, what does "E" mean in equation (7)? I guess "A".
Response: We agreed with respected reviewer comment. Now in revised manuscript, we made correction in Equation 7 by replacing “E” with “A”, nevertheless, due to the movement of equations in computational procedure, the numbering of Equation is change from 7 to 1. Thanks for the comment.
Query 3 In Table 2, the C21 LP(1) ---> N6-C19 (PI*) charge transfer of the HMBA compound is striking, it is rare for this carbon to have an unshared pair, however it is not very clear whether C21 corresponds to C5 or C9 in Figure 1, in any case, it deserves an explanation.
Response: Now in revised manuscript, we critically rechecked the values and made correction in Table 2. Thanks for the comment.
Query 4. When I compare the orbital energies in Table 3 with the values in Table 4, I come to the conclusion that Table 4 is in a.u. instead of eV as the table text says. In any case, the softness values should be in eV-1 instead of eV.
Response: Thanks for the comment. Now in revised manuscript, we rechecked the above mentioned values and made correction. Now the values in Table 4 are in eV while softness is in eV-1 according to the suggestion of respected reviewer.
Query 5. The heading of Table 2 is confusing E(J)E(i)b is repeated twice, I think it should be like Table S6. The units of E(2) in kJ/mol in Tables S6 and S7 I think should be kcal/mol.
Response: Thanks for the comment. Now in revised manuscript, we made correction in the heading of Table 2 according to the suggestion of respected reviewer. The units of energy also corrected in Table S6 and S7 (E is in kcal/mol).
Query 6. The text just before Table 2: "The main transition between occupied and unoccupied orbitals of HMBA and DHBA compounds are illustrated in Table 2" is poorly worded, as it is talking about partial charge transfers, not electronic transits.
Response: Thanks for the comment. Now in revised manuscript, we made improvement in the above mentioned text as
“Some important outcomes of NBOs for compounds HMBA and DHBA are listed in Table 2 while, Tables S6 and S7 demonstrated rest of the NBO transitions representing established conjugation.”
Query 7. I think the atomic numbering in Fig. 1 does not correspond to Tables 2, S6 and S7, there is no way to follow those tables.
Response: Now in revised manuscript, we set the atomic numbering in Tables 2, S6 and S7 according to Figure 1. Thanks for the comment.
Query 8. The section "molecular geometric optimization" is very difficult to follow as it is, it would be much better as a graph of the linear regression between the data with its corresponding regression equation and R2 value and commenting only on some particularly noteworthy cases. Furthermore, the information it provides is poor for it to be a section, it would be better if it were included in the section "Description of SCXRD analysis of HMBA and DHBA.
Response: Thanks for the suggestion. Now in revised manuscript, we removed the "molecular geometric optimization" heading and developed graphs for comparative analysis between SCXRD and DFT values according to respective reviewer suggestion and shown in Figure S5
Query 9. In the "FMO analysis" section, where is the TD-DFT analysis? In the Computational procedure it says: "Time-dependent density functional theory (TDDFT) was used to study electronic transitions: FMOs and MEP" and this is the section where it is logical to put it. It would also be good if it were accompanied by the corresponding experimental UV-Vis spectra for a proper comparison.
Response: Now in revised manuscript, we mentioned the TD-DFT in FMOs and MEP analyses according to the suggestion of respected reviewer. Nevertheless, we didn’t performed UV-Vis investigations for current crystals therefore didn’t made comparative analyses between FMOs and UV-Vis. Thanks for the comment.
Query 10. The section "Natural population analysis" as it is presented does not make much sense if it cannot be related to the Second Order Perturbation Theory Analysis (what the authors have called NBO analysis) to complete the information provided by the charge transfers, it should be included in that section. It should also be related to the section on Hirshfeld surfaces.
Response: We are thankful to the respected reviewer for in-depth analysis of our manuscript. The mulliken charges discussion under the heading of “natural population analysis” are taken from NBO files. We also developed relation between Hirshfeld surfaces and NPA charges. Thanks for the suggestion.
Query 11. The section "Molecular electrostatic potential" has the same comment as the section "Natural population analysis", it should be used to complete the information in the section "Natural Population Analysis".
Response: Thanks for the suggestion. Now in revised manuscript, we discussed the "Molecular electrostatic potential" along with "Natural population analysis" according to the suggestion of respected reviewer.
- Results and Discussion Section:
Query 12. In the pdf that I downloaded from the journal there are 8 sections 1.1 three sections 1. and several sections 1.1.1. I imagine that this has been produced when generating the pdf file with the journal's application. To avoid this type of formatting error, I usually enter the numbering "by hand" instead of letting the editor handle it automatically
Response: Thanks for your suggestion. Now in revised manuscript we made correction in formatting.
Round 2
Reviewer 1 Report
No comments.
Reviewer 2 Report
accept.